# The Prokineticin System in Inflammatory Bowel Diseases: A Clinical and Preclinical Overview

**DOI:** 10.3390/biomedicines11112985

**Published:** 2023-11-06

**Authors:** Giada Amodeo, Silvia Franchi, Giulia Galimberti, Benedetta Riboldi, Paola Sacerdote

**Affiliations:** Dipartimento di Scienze Farmacologiche e Biomolecolari “Rodolfo Paoletti”, University of Milan, Via Vanvitelli 32, 20129 Milan, Italy; silvia.franchi@unimi.it (S.F.); giulia.galimberti@unimi.it (G.G.); benedetta.riboldi@unimi.it (B.R.); paola.sacerdote@unimi.it (P.S.)

**Keywords:** prokineticin system, inflammatory bowel diseases (IBDs), Crohn’s disease (CD), ulcerative colitis (UC), chronic pain

## Abstract

Inflammatory bowel disease (IBD) includes Crohn’s disease (CD) and ulcerative colitis (UC), which are characterized by chronic inflammation of the gastrointestinal (GI) tract. IBDs clinical manifestations are heterogeneous and characterized by a chronic relapsing-remitting course. Typical gastrointestinal signs and symptoms include diarrhea, GI bleeding, weight loss, and abdominal pain. Moreover, the presence of pain often manifests in the remitting disease phase. As a result, patients report a further reduction in life quality. Despite the scientific advances implemented in the last two decades and the therapies aimed at inducing or maintaining IBDs in a remissive condition, to date, their pathophysiology still remains unknown. In this scenario, the importance of identifying a common and effective therapeutic target for both digestive symptoms and pain remains a priority. Recent clinical and preclinical studies have reported the prokineticin system (PKS) as an emerging therapeutic target for IBDs. PKS alterations are likely to play a role in IBDs at multiple levels, such as in intestinal motility, local inflammation, ulceration processes, localized abdominal and visceral pain, as well as central nervous system sensitization, leading to the development of chronic and widespread pain. This narrative review summarized the evidence about the involvement of the PKS in IBD and discussed its potential as a druggable target.

## 1. Introduction

Inflammatory bowel diseases (IBDs) are a group of disorders characterized by chronic inflammation of the gastrointestinal (GI) tract, including Crohn’s disease (CD) and ulcerative colitis (UC). The two diseases only differ in lesion position and depth along the gastrointestinal tract. Indeed, CD can affect any portion of the GI tract (from the mouth to the anus), causing a transmural inflammation that reaches multiple layers of the GI tract’s walls, while UC essentially involves the colon and rectum, with inflammation limited to the mucosal layer (Figure 1). However, sometimes, the two diseases can be indistinguishable, and in these cases the condition is defined as unclassified inflammatory bowel disease [1]. It has been estimated that there are more than 7 million IBD patients worldwide. In the West, although the IBDs’ incidence has stabilized, the prevalence is steadily increasing due to the rise in patient survival; instead, in newly industrialized countries, IBDs’ incidence is rapidly increasing [2,3]. IBDs affect people of both genders and all ages, with a higher percentage between the ages of 15 and 40 [4]. The IBDs’ onset in adolescents/young adults results in longer and potentially more complex disease duration during adulthood and old age, making it difficult to manage in an aging population. Both IBDs are characterized by a relapsing-remitting course and share common symptoms, such as diarrhea, fatigue, abdominal pain, cramps, stool blood, decreased appetite, and unintentional weight loss [5,6]. 

Among these symptoms, the most relevant and disabling one for IBD patients is abdominal pain. During the active phase of the disease, most IBD patients experience pain, which usually should improve/resolve within the decrease in disease activity (the remission phase). However, a significant percentage of IBD patients continue to feel painful symptoms, despite both inflammation resolution and clinical remission achievement [7]. This occurs as during chronic inflammatory states, sensory sensitization pathways lead to persistent changes in afferent neurons and pain processing circuits in the central nervous system [8]. In this way, the persistent pain typical of IBD patients is not a simple result of sensory input. In addition, pain processing and sensory pathways are also modulated via emotional and cognitive factors. In addition, a high prevalence of neuropsychiatric comorbidities (like anxiety and depression) has been observed in IBD patients, which could significantly contribute to the clinical manifestations of chronic pain [9,10]. Therefore, the identification of a novel target/pathway that can modulate inflammation, decrease the excitability of sensitized afferent pathways, and regulate emotional and/or cognitive functions appears essential in IBD resolution. 

In the IBDs, the dysregulation of both innate and adaptive immune responses is present [11]. Innate intestinal immunity is a first-line defense able to induce a rapid and non-specific reaction to pathogens and/or excessive intestinal microorganisms’ entry. Changes in this balance, implemented by neutrophils, monocytes, macrophages, dendritic cells, lymphoid cells, and natural killer cells, are related to the intestinal inflammation typical of IBDs. Although the exact etiology of IBDs still remains unknown, it appears clear that in predisposed individuals, an unresolvable mucosal damage is an important disease feature [12,13]. The cause of this early damage could be related to infectious agents, chemical compounds, or metabolic alterations (possibly related to diet-mediated dysbiosis), and the subsequent disease perpetuation could be due to the unsuccessful resolution of the inflammatory response to this initial lesion [14,15,16,17]. The cells of the innate immunity respond to the insult by producing cytokines and chemokines, thus first activating the complement cascade and phagocytosis, and subsequently the adaptive immune response, supporting in this way an inflammatory condition [18]. Given the striking similarities in the clinical and morphological manifestations of ulcerative colitis and Crohn’s disease, it has been hypothesized that their trigger could be induced by soluble mediators that are generated during an inflammatory process, where, therefore, the immunological perturbations associated with IBD are causes rather than effects.

Against this backdrop, a new family of chemokines, the prokineticin system (PKS), has attracted increasing interest. Indeed, the PKS appears to be involved in IBDs at multiple levels and could represent a key mechanism in the first phases of immune dysregulation typical of IBDs, as well as in supporting its symptoms and comorbidities. Therefore, in this review, we discussed the involvement of the PKS in intestinal motility, inflammation, angiogenesis, tumorigenesis, and pain, presenting preclinical and clinical evidence of the involvement of the PKS in IBDs.

## 2. Methodology

A search was conducted on May 2023, according to the following criteria:No filter time.Database: PubMed, Cochrane Library, and Embase.Language: only full-text in English.Keywords: terms used were “prokineticin system” OR “prokineticin 1” OR “prokineticin 2” OR “prokineticin receptors” combined with “inflammatory bowel disease” OR “Crohn’s Disease” OR “ulcerative colitis” through the Boolean operator AND.Key terms were searched in the title, abstract, and keywords of the papers. The search was open to all parameters in order to avoid information loss. MeSH terms were not used.The papers’ selection was based on a critical reading.

The corresponding author revised the titles and abstracts of all papers. Abstracts of the selected articles were read to determine whether they fulfilled the inclusion criteria. After the first screening, 22 papers were selected. Furthermore, some papers were selected from the references of the selected manuscripts and from private bibliographic lists of the authors. A total of 106 papers were included in this narrative review.

## 3. The Prokineticin System: A General Overview

The prokineticin system (PKS) includes two ligands, namely prokineticin 1 (PK1 or EG-VEGF (endocrine gland derived-vascular endothelial growth factor)) and prokineticin 2 (PK2 or Bv8), as well as their receptors PKR1 and PKR2 (Figure 2). Both prokineticins (PKs) are small secretory proteins (about 10 kDa) classified as novel chemokine-like peptides based on their sequence, structure, and function. PKs are highly conserved among species, from invertebrates to humans. The first peptides were isolated from the skin secretion of the frog *Bombina variegata* (called Bv8) [19] and from the black mamba snake venom as a non-toxic compound (called mamba intestinal toxin 1, MIT-1, or venom protein A (VPRA)) [20,21]. 

Subsequently, the DNA sequences encoding for MIT-1 and BV8 were used as probes to investigate whether orthologues were also present in the human genome; in this way, the bioactive peptides PK1 (86 aa) and PK2 (81 aa) were identified [22]. PKs exhibit highly conserved structural homologies among species, essential for their bioactivity, consisting of a carboxy-terminal cysteine-rich motif forming five disulfide bridges, a Trp residue at position 24, and a N-terminal AVITGA sequence (alanine, valine, isoleucine, threonine, and glycine), essential for the correct binding to their receptors [23,24,25]. The receptors of these two prokineticins, PKR1 and PKR2 (PKRs), are G protein-coupled receptors and possess 85% amino acid identity, mainly only differing in the N-terminal region [26,27]. PKRs can couple to Gq, Gi, and Gs, thus activating different intracellular signaling pathways depending on the cell type that expresses them and their location. Since the PK human orthologues’ identification, many efforts have been undertaken to identify which tissues and organs express PKS members. An important contribution was made by Zhou’s research group, who first probed a human master blot using human prokineticin cDNA fragments [22], and then investigated the tissue distribution of PKRs as mRNAs via reverse transcription PCR analysis on a panel of tissue cDNAs [27] (Figure 3). Nowadays, it is known that both PKs and PKRs are widely expressed in different organs and tissues (Table 1) and are involved in a plethora of biological effects, such as intestinal motility, neurogenesis, angiogenesis, circadian rhythms, hematopoiesis, sensory processing, and nociceptive signaling, as well as being an important player in the pathophysiology of inflammation and pain [28,29,30,31,32,33,34]. Furthermore, emerging evidence has also indicated the PKS’s involvement in the pathophysiology of different diseases that affect the nervous and reproductive systems, as well as myocardial infarction and tumorigenesis [35,36,37,38]. 

However, to date, a comprehensive multi-organ comparative study, in which it is possible to verify the quantitative difference in PKS levels in different organs, has only been conducted by Zhou’s group [22,27]; therefore, it still remains difficult to identify quantitative differences between tissues, since PKS expression analysis in different tissues and organs has often been performed by different authors with different methodologies (mRNA, immunohistochemistry, Western blotting, etc.). However, we can state, based on our recent study [39], that the mRNA levels of the PKS in the colon and in the nervous system (myenteric plexus, dorsal root ganglia, and spinal cord) were similar, demonstrating the relevance of the PKS in both inflammation and pain processing. We believe that detailed multi-organ characterization studies should be conducted to offer a comprehensive overview of PKS expression. Furthermore, performing a characterization of the PKS in different mammals would offer a clearer distribution of the PKS, making also the possibility of performing quantitative comparisons between these different species. This could be an important step for translating the results obtained in preclinical studies into the clinic.

## 4. The Role of the PKS in Inflammatory Bowel Disease

### 4.1. Evidence of the PKS in Gastrointestinal Motility, Contraction, and Secretion

In IBD patients, the presence of gastrointestinal motility disorders can appear in several forms, like dyspepsia, diarrhea, urgency, fecal incontinence, and/or constipation [40,41]. These disorders are frequent; indeed, they afflict more than a third of IBD patients, significantly compromising their overall quality of life [42]. Recent evidence has suggested that intestinal dysmotility and IBDs share a common link, i.e., inflammation, with overlapping symptoms and predisposing factors, which are not always related to mucosal damage [43,44,45]. In addition, in IBD patients with a dysfunctional gut, pathological abnormalities involving the enteric nervous system are known [46,47]. In Figure 4, a schematic representation of the gastrointestinal tract and its innervations is reported. Nowadays, it is known that, under physiological conditions, all PKS members are extensively expressed in the gastrointestinal tract, including the enteric nervous system (Figure 5). In this context, the PKS has been shown to be largely involved in the contractions and secretion processes of the gastrointestinal tract, where PKS alterations could therefore be the cause or con-cause of disturbances. In this regard, Schweirz and colleagues [20] were the first to investigate the prokinetic role of mamba intestinal toxin (MIT-1) through ex vivo experiments in guinea pig gastrointestinal tract tissues. These authors observed that the administration of MIT-1 induced a mighty contraction of both longitudinal ileal muscle and the distal colon, as well as relaxation of the proximal colon. Furthermore, in order to identify the target of MIT-1 at the gastrointestinal tract level, and in particular to evaluate whether the action was pre- or post-synaptic, the effects of tetrodotoxin (TTX, capable of blocking the propagation of the action potential) were studied. TTX counteracted both the relaxing effect of the proximal colon and the contractile effect of the distal colon induced by MIT-1, thus suggesting that MIT-1’s action occurred at the pre-synaptic level. The authors in this article also hypothesized the possible existence of an endogenous compound in humans, whose identification would have represented an important discovery for both intestinal motility control and intestinal system pathophysiology. Indeed, in the following years, Zhou’s group was first able to identify two homologues in humans, namely prokineticin 1 and prokineticin 2 (PKs) [22], and shortly after their binding sites (PKR1 and PKR2) [27]. Indeed, it was this same research group that called the prokineticin system in this way, due to the specific and potent contractile activity of these peptides on the gastrointestinal tract smooth muscle. In these studies, the effects of both recombinant PKs were evaluated on smooth muscle preparations isolated from the guinea pig. Both prokineticins potently stimulated ileal longitudinal muscle contraction, but only PK1 was able to stimulate the contraction of both fundic muscle strip and the proximal colon. Moreover, no effect of PK1 was observed on other smooth muscle tissues, including the aorta, femoral artery, trachea, and gallbladder. 

Therefore, as also suggested by Schweirz et al. [20], the contractile effect of prokineticins seems to be specific for the gastrointestinal smooth muscle, suggesting that the PKS could represent an important endogenous regulator of gastrointestinal motility. In subsequent studies, the same research group identified that PKR1 is more abundantly expressed in the gastrointestinal tract than PKR2, suggesting that PKR1 may be the receptor mediating PKs’ function in the gastrointestinal system [27]. In the following years, other research groups also began studies about the PKS’s role in gastrointestinal contraction, producing, however, conflicting results. A preliminary report suggested that PK2 was able to increase the emptying of a liquid meal from the rat stomach [48], thus suggesting an involvement of PK2 in gastrointestinal motility/contraction. However, a final publication on this study is not present in the literature. Bassil and colleagues [49], instead, investigated the contractile capacity of PK2 in isolated preparations of mouse stomach, intestine, and colon. Their hypothesis was that PK2 acts through cholinergic activation, rather than directly on smooth muscle. The authors concluded that PK2 was unable to induce contractions. Therefore, these results are in contrast to both the preliminary report [48] and the study by Lin et al. [27]. However, these divergences could depend on both different animal species and the tissue isolation/processing method used. Additionally, it could be hypothesized that PK2’s activity could be mediated outside the enteric nervous system. This hypothesis is supported by the widespread expression of PK2 in brain areas such as the nucleus tractus solitarius, which communicates with the gastrointestinal tract via the vagus nerve. It must be emphasized that all these studies have been conducted in isolated organs, where, therefore, the complexity of a complete organism is lacking.

**Figure 5 biomedicines-11-02985-f005:**
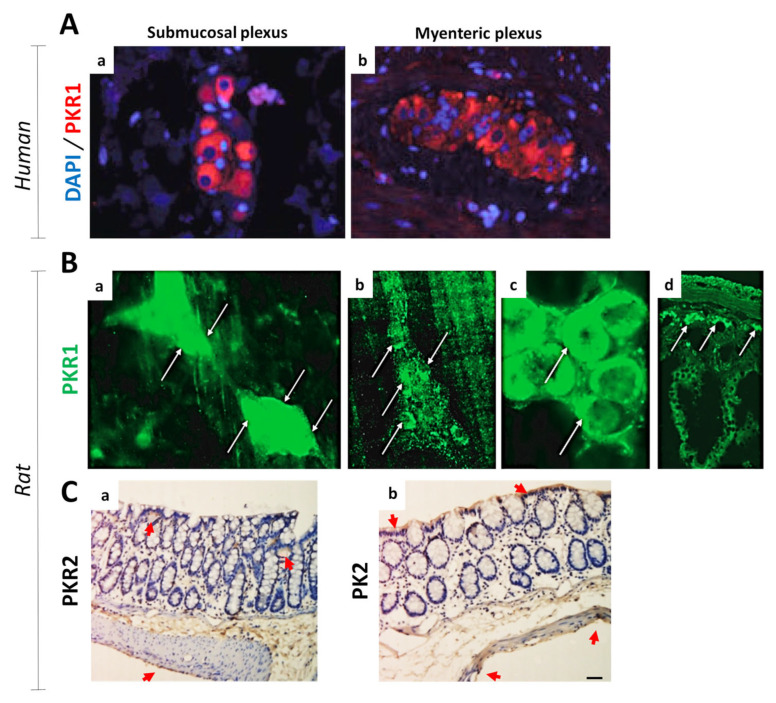
**PKRs and PK2 expression in the gastrointestinal tract**. (**A**) Immunohistochemical analysis of PKR1 expression in the (**a**) myenteric and (**b**) submucosal plexus of the human gastrointestinal tract (in blue, DAPI; in red, PKR1). (**B**) PKR1 immunoreactivity in rat ileum. In detail (see white arrows), PKR1 was detected: (**a**) in the majority of submucosal neurons, (**b**) in some neurons and fibers in the myenteric plexus, (**c**) in the epithelial cells at the base of the crypts primarily, and (**d**) in the cell basolateral regions. (**C**) Immunohistochemical evaluations in the rat colon of (**a**) PKR2 and (**b**) PK2. Both PKR2- and PK2-positive expression (yellow stains, see red arrowheads) was mainly detected in the mucosal epithelium, proximal glands of the mucosal epithelium, interstitium, and colon muscle layer. These images were modified and republished, in accordance with specific editor copyright, by (**A**) Watson et al. [50]; (**B**) Wade et al. [51]; and (**C**) Zhou et al. [52].

Subsequently, Hoogerwerf et al. [53] confirmed the presence of PK1 and PKR1 in the mouse distal colon and described PKR1’s expression in the myenteric plexus at neuron level. In their studies, as already observed by Zhou’s group [22,27], PK1 reduced the frequency and amplitude of spontaneous contraction in the proximal colon, thereby causing an increase in contraction, but not in the distal colon. This dissimilarity between the two colon portions has been attributed to the fact that the myenteric plexus of the proximal colon, having more NOS-containing cells, has greater NOS activity than the distal colon, resulting in greater NO-mediated non-adrenergic and non-cholinergic relaxation in the proximal colon. Indeed, in this study it was identified that myenteric plexus neurons expressing PKR1 co-localize with a small subset of nNOS neurons (Figure 6). The authors also demonstrated that the PK1-treated colon released significantly higher concentrations of NO than the controls. In a preliminary report, Wade et al. [51] observed, in ex vivo studies in rat small intestine, that PK1 treatment stimulated fluid accumulation and accelerated transit through neuronal and non-neuronal pathways. Furthermore, in vivo treatment with PK1 stimulated a dose-dependent secretory response, thus revealing the potent secretogenic activity of PK1. Of great interest are the results obtained from the pretreatment of intestinal tissue with both a prostaglandin (PG) EP4 receptor antagonist and a cyclooxygenase-1 inhibitor. Indeed, the authors demonstrated, for the first time, that these pretreatments block PK1’s activity in the gastrointestinal tract, thus suggesting that at the gastrointestinal level PK1’s effect is mediated via PGE activity. Subsequent preclinical studies by Ngan and colleagues [54,55,56] found that PK1, but not PK2, not only mediates muscle contraction of the mature gastrointestinal tract, but also regulates enteric nervous system formation during embryonic development. In mammals, the enteric nervous system arises from neural crest cells that enter the foregut and colonize the entire wall of the gastrointestinal tract. The authors observed that neuronal crest cells, in addition to PK1, also express PKR1 (Figure 7). 

Experiments with PKR1 knock-out (KO) mice confirmed that, during ENS development, PK1 only exhibits its functions via PKR1, and not PKR2. However, although PKR1 KO mice exhibit morphological or physiological deficits, these are not as severe as those associated with aganglionosis present in *Ret*^−/−^ or *Gdnf*^−/−^ mice. This indicates that a functional enteric nervous system may develop even without PKR1 expression. In vitro studies on murine portions of the embryonic gastrointestinal tract established that PK1 is able to promote the proliferation and differentiation of neuronal crest cells into neural/glial lineages (Figure 7). Therefore, it has been demonstrated that the PK1/PKR1 activation contributes to myenteric and submucosal plexus formation. Additionally, it was observed that PK1 also induces the expression of inhibitory neurotransmitters, such as nitric oxidase (NO) and vasoactive intestinal peptide (VIP), further suggesting that this ligand mediates the formation of inhibitory motor neuron pools. In a subsequent publication, Wade et al. [57] evaluated the expression levels of PKS members in rat gastrointestinal tract portions (the fundus, pylorus, duodenum, jejunum, and ileum), identifying a differential expression. In general, in all portions, a higher expression level of PK1 than PK2, and of PKR1 than PKR2 (especially in the ileum), was observed. 

**Figure 7 biomedicines-11-02985-f007:**
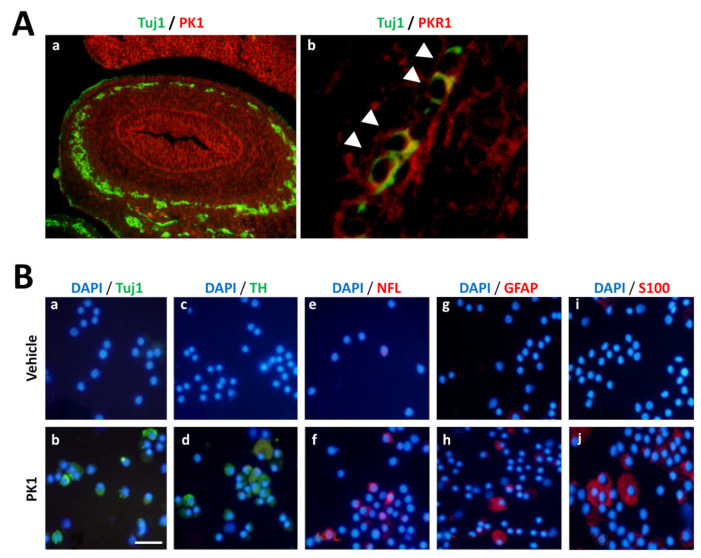
**PK1 and PKR1 in enteric neural crest cells.** (**A**) Representative immunohistochemical images of PK1 ((**a**), in red) or PKR1 ((**b**), in red) and Tuj/1 ((**a**,**b**), in green) localization in mouse embryonic gut at 15.5 (**a**) and 13.5 (**b**) days, respectively. The enteric neural crest cells (NCCs), marked with Tuj/1, are localized in the gut mesenchyme and colocalized (arrowhead) with PKR1. PK1 is localized at the mucosa and mesenchyme. (**B**) NCCs cultured with vehicle or PK1 for 48 h. NCCs supplemented with PK1 showed a marked neuronal ((**a**–**d**), in green; (**e**,**f**), in red) and glial expression ((**g**–**j**), in red) in comparison with the vehicle. Neuronal markers: Tuj1 (class III beta-tubulin), TH (tyrosine hydroxylase), and NFL (neurofilament light); glial markers: GFAP (glial fibrillary acidic protein) and S100. These images have been edited and republished, in accordance with the specific editor copyright, by Ngan et al. [54].

Ex vivo treatment of the gastrointestinal tract with PK1 evoked contractions of the duodenum, jejunum, and distal ileum, as well as relaxation of the distal colon. However, the maximal contractile responses to PK1 were obtained in ileal tissues. On the other hand, in vivo experiments have shown that oral administration of PK1 causes both an acceleration of upper gastrointestinal transit and an increase in the luminal fluid content in the intestine, suggesting that in the gut, PKS may act via local paracrine mechanisms to evoke potent prokinetic and secretory effects. 

Altogether, these data suggest a clear involvement of the PKS in gastrointestinal motility/secretion, where the key players appear to be PK1 and PKR1. However, further studies are needed to explore the PK1/PKR1-mediated pathophysiological responses of IBDs in the gastrointestinal tract.

### 4.2. Evidence of the PKS in Angiogenesis and Inflammation

As known, IBDs are characterized by pathological intestinal inflammation associated with epithelial damage (like ulcers, fistulas, strictures, and abscesses) [58]. In these diseases, angiogenesis is an important component for both pathogenesis and inflammation [59]. Additionally, cytokines and chemokines are directly implicated in IBD onset and have a crucial role not only in intestinal inflammation control but also in damage and regeneration processes. Recent studies have shown that aberrant and excessive cytokine responses characterized the initial phase of IBDs, causing acute or subclinical inflammation. In predisposed subjects, these conditions fail to be resolved, leading to chronic intestinal inflammation development due to the uncontrolled activation of the immune system (crucial role of macrophages and T cells) [60]. Moreover, during IBD onset, physiological angiogenesis switches into a pathological one [61]. Among the factor(s) driving inflammation and angiogenesis, an important role could be played by the PKS. The Ferrara group [62,63] identified PK1 as a selective angiogenic mitogen for endocrine gland-endothelial cells. Indeed, this protein presented striking biological/functional similarities to the VEGF family (but no structural homology), and for this reason the authors proposed its designation as endocrine-derived vascular endothelial growth factor (EG-VEGF). It has been demonstrated that VEGF and EG-VEGF can function in a coordinated and/or complementary manner to regulate angiogenesis and permeability (Figure 8). 

Subsequently, the same research group characterized PK2 and PKRs’ distribution in different hematopoietic cell types, and also evaluated, via in vivo and in vitro experiments, the PK1 and PK2 treatment effect on hematopoiesis and hematopoietic cell mobilization [33]. Their results revealed that, in humans, PK2 is expressed by hematopoietic stem cells, bone marrow-derived mononuclear cells, and innate immune system cells (in particular, dendritic cells, monocytes, and neutrophils). Meanwhile, PKR1 was expressed in monocytes, CD8+ lymphocytes, and hematopoietic stem cells, while PKR2 was expressed in CD8+ and CD4+ lymphocytes, neutrophils, monocytes, and B cells, as well as hematopoietic stem cells. These authors also demonstrated that mouse hematopoietic stem cells express PK2 and PKRs. Moreover, through chemotaxis assays with human monocytes, it has been demonstrated that both PK1 and PK2 can induce a significant dose-dependent increase in the number of migrating monocytes. However, this effect was more significant for PK2, thus demonstrating, for the first time, its monocyte chemoattractant activity [33]. The authors also observed that neutrophils infiltrating inflammation sites expressed high PK2 levels, leading to a significant overall increase in PK2 in inflamed versus healthy tissue. Additionally, the authors observed that in mice both PK1 and PK2 treatments significantly increased the number of circulating leukocytes, monocytes, and neutrophils. Therefore, this study demonstrated that PK2 is able to modulate innate and adaptive immune system cells, probably through autocrine or paracrine signaling mechanisms. In addition, further studies have shown that PK1 is involved in monocyte differentiation and function [64]. Indeed, it was discovered through in vitro studies (on human and mouse cells) that although the PK1 effect on cell proliferation was minimal, its effect on monocyte/macrophage lineage differentiation was substantial. The authors also observed that immune cells only stimulated in vitro with PK1 did not increase pro-/anti-inflammatory cytokine production. However, a PK1/LPS co-stimulation revealed the PK1 pro-inflammatory activity. Indeed, it was able to considerably increase pro-inflammatory cytokine production and suppress the anti-inflammatory one, compared to culture with LPS alone. These data indicated that PK1 activates monocyte differentiation, inducing a pro-inflammatory Th1 response. This suggests that PK1 and PK2 may be involved in immune response regulation by altering monocyte differentiation and activation [64].

**Figure 8 biomedicines-11-02985-f008:**
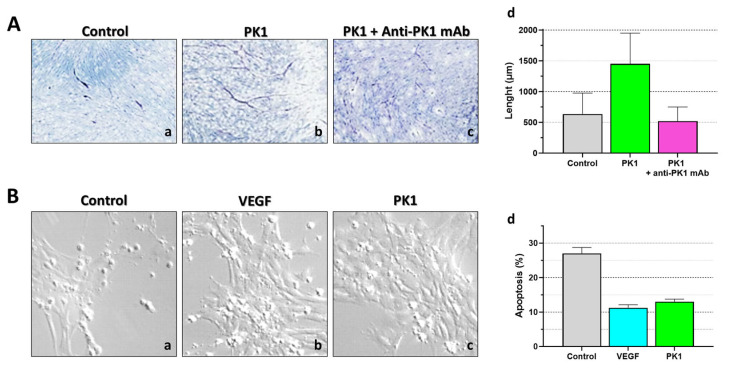
**PK1’s effect on endothelial cells in angiogenesis**. (**A**) Capillary tube formation (stained tube-like structures in purple) was evaluated, using a specific angiogenesis kit (Kurabo Co), (**a**) in a standard cultured medium condition, (**b**) supplemented with PK1 (10 ng/mL), or (**c**) supplemented with both PK1 (10 ng/mL) and anti-PK1 mAb. (**d**) For each condition, capillary tube length was analyzed/quantified using the MacSCOPE program. The PK1-supplemented culture markedly increases the capillary tubes’ length, thereby stimulating angiogenesis, whereas, if the effect of PK1 is blocked, angiogenesis is also arrested. (**B**) Adrenal cortex-derived endothelial cells were cultured in (**a**) serum-free medium with/without (**b**) VEGF (0.13 nM) or (**c**) PK1 (50 nM). (**d**) Quantification of apoptotic cells using the fluorescence-activated cell sorting assay. After 24 h, starved endothelial cells showed 30% apoptosis, while VEGF or PK1 supplementation was able to rescue cells from this doom, indicating that PK1, similarly to VEGF, promotes endothelial cell survival. These images/graphs have been edited and republished, in accordance with the specific editor copyright, by Nakazawa et al. [65] and (B) Lin et al. [63].

On this line of thought, our research group focused on the PK2 immunomodulatory properties [66]. Briefly, via chemotaxis assays with murine peritoneal macrophages, we observed that PK2 (at very low concentrations, phentomols) promotes substantial macrophage migration. Furthermore, we demonstrated, in the same cell population, the development of the pro-inflammatory phenotype induced by PK2. Indeed, in cultures of murine macrophages supplemented with PK2, there was a marked increase in pro-inflammatory cytokines and a reduction in IL-10 production. Of great interest were our data resulting from PKR1-deficient mice. Using peritoneal macrophages obtained from PKR1 KO mice, we observed that both chemotactic activity and cytokines’ modulation by PK2 were completely abolished. Our results demonstrated that the effects exerted by PK2 on macrophages were mediated via the PKR1 receptor. Our data also suggested the involvement of a G_q_ protein in the observed effects. Subsequently, we also evaluated, via in vivo and in vitro studies, the effect of PK2 treatment on cytokine production by lymphocytes isolated from the spleen of both WT and PKR1 KO mice [67]. Our in vitro experiments demonstrated that PK2, in combination with the mitogen concavalin-A, significantly decreased the anti-inflammatory Th2 cytokines, IL-4 and IL-10, without affecting Th1 cytokines’ levels. Similar results were also obtained when PK2 was administered in vivo. Also, in this case, the effects were mediated by PKR1, since in PKR1 KO mice the modulation by PK2 was not present. Our results suggested that PK2 was able to shift the Th1/Th2 balance towards a Th1 pro-inflammatory phenotype, and that this effect was mediated via the PK2/PKR1 interaction.

All these data could lead us to speculate that in predisposed subjects, a low inflammation of the gastrointestinal tract may lead to a significant increase/exacerbation in PKS members. This upregulation, activating the immune system’s response, induces the recall, in loco, of immune cells from several parts of the body, creating a vicious circle that fuels intestinal inflammation.

### 4.3. Evidence of the PKS in Gastrointestinal Tumorigenesis in IBDs

Colorectal cancer (CRC) is the third most common tumor and the fourth most ordinary cancer mortality cause worldwide [68]. It is now known that IBDs are associated with a greater risk of developing CRC than the general population (about a 3-fold increase). Moreover, CRC represents one of the main causes of IBD patient death [69,70]. It has also been estimated that CRC onset in IBD patients has a worse prognosis than in patients with no IBD history [71]. Although the exact etiology and pathogenesis of this type of cancer are not yet fully understood, recent preclinical and clinical advances have indicated that the long-standing chronic inflammation is a driver for neoplastic progression [72,73]. Indeed, innate and adaptive immune system cells play an important role in cancer onset, where the crosstalk between these cells mainly occurs through a network of cytokines that guide and maintain inflammation and contribute to tumorigenesis through, for example, oxidative stress, epithelial cell proliferation, and angiogenesis [74]. In this context, it has been shown that the PKS is associated with malignant carcinoma development [36]. In detail, PK1, an already known angiogenic factor, has been associated with tumorigenesis, cellular invasion, and metastases onset in several tumors [75,76,77,78], including CRC [65,79]. From these studies, which analyzed PK1 expression in human colorectal cancer tissue samples, it emerged that PK1 upregulation was present in about 40% of the CRC samples collected. Furthermore, it was demonstrated that the frequency of PK1 overexpression was significantly higher in cases with serous, lymphatic, and venous invasion, metastasis formation, and higher-stage disease [65,79]. Additionally, the recurrence rate and prognosis for patients with PK1-positive primary tumors were significantly worse and associated with a poorer survival rate. Therefore, it has been suggested that in colorectal cancers, PK1 functions as an important angiogenetic factor in both primary lesion and metastasis formation and may therefore be considered a possible CRC prognostic factor. A recent study demonstrated that plasma PK1 level may also be a potential predictive marker, especially in patients with advanced-stage CRC [80] (Figure 9). Also, PK2’s role in CRC has been recently studied [81]. In a murine preclinical study, CRC cells transfected with the PK2 gene were implanted into mice. This induced a significant increase in the cancer mass size and in the number of CD31^+^ cells compared to animals treated with CRC cells transfected with the empty vector (without PK2). Furthermore, when the same animals were treated with PK2 siRNA, a significant reduction in tumor mass size was observed. Subsequently, it was observed that in several human CRC cell lines a significant increase in PK2 expression levels was present, and PK2 upregulation was shown to play a role in cancer recurrence and progression [82]. In this study, 436 patients with CRC (stages I–III) were included, and PK2 expression in the primary tumor was studied. Marked PK2 upregulation was detected in 51% of CRC patients, and was significantly associated with lymphatic invasion, metastasis, and increased recurrence rates.

These data demonstrate that the PKS may be directly involved in CRC development and progression, and suggest that both ligands, PK1 and PK2, may work synergistically. PK1, functioning as an angiogenic factor, is involved in the formation of both primary lesions and metastases, while PK2 fuels the growth/volume of cancer masses and plays a role in recurrence. 

### 4.4. Evidence of the PKS in Pain: Focus on IBD Model

Several studies have attempted to investigate the PKS’s involvement in nociception and in pain development and maintenance [29]. Through in vivo and in vitro preclinical experiments, it has been demonstrated that PK2 stimulation/injection induces PKRs’ activation in primary afferent C fibers, sensitizing the nociceptors to thermal, mechanical, and chemical stimuli [19,83,84,85,86]. Subsequent studies supported the PKS’s physiological role in nociception, as PKR and PK2 KO mice were less sensitive to noxious stimuli than wild-type (WT) mice, with less hyperalgesia onset after tissue damage [87,88,89,90]. A role of the PKS has also been demonstrated in the development and maintenance of both chronic inflammatory [91,92] and neuropathic pain of various etiologies [93,94,95,96,97,98,99,100,101]. In these studies, it has been well demonstrated that the presence of aberrant pain was associated with an upregulation in PK2 and PKRs, and that selective PKS antagonism was able to counteract pain symptoms and neuroinflammation. Studies that investigated IBD pain and the PKS in IBD models are also present in the literature. Kimball et al. [102] was the first manuscript that documented significant increases in PK1 and PKR1 in an IBD model. In this study, a mouse model of acute colitis induced via intracolonic administration of mustard oil (MO) was used. This compound induces visceral allodynia and hyperalgesia, directly stimulates small intestinal nerve fibers, and causes an acute inflammatory state. The authors observed that in the acute time window (2–6 h post-MO) mouse colons presented severe inflammatory damage, resulting in edema formation and weight gain, as well as diarrhea presence. These alterations were also associated with a significant increase in PK1 and PKR1. At subsequent observation points (24–72 post MO) inflammatory damage was further increased, while PK1 and PKR1 levels returned to physiological levels. However, it is interesting to note that a significant increase in IL-1β and IL-6 was detected at this time (24 h post-MO). This suggests that in intestinal inflammatory pathways PK1/PKR1 might be a possible trigger, promoting, in loco, the recruitment of immune cells and inducing pro-inflammatory cytokine production. Subsequently, the role of the PKS in visceral pain was also investigated by Wu’s research group [52,103]. A model of visceral hypersensitivity induced via chronic and repetitive colorectal distension (CCD) in rats was used in both of these studies. The CCD model induces sustained and prolonged visceral hypersensitivity, allowing for a chronic phase of IBD pain to be studied. The authors observed that in this model a significant increase in visceral hyperalgesia correlated with an increase in both PKs and PKRs at the colon level. The expression of PKS members was detected in different districts of the colon (see Table 1). The authors demonstrated that the therapeutic treatment with moxibustion was able to attenuate visceral hyperalgesia, also positively modulating PKS levels. Although no molecular mechanisms have been suggested, these data demonstrate that PKS upregulation is related to visceral hypersensitivity onset.

Watson and colleagues [50], using several different IBD models, investigated PK2’s role in visceral pain and evaluated the effect of a selective PKR antagonist (Compound 3). Initially, in vitro experiments were performed on rat cultures of both ileal myenteric neurons and DRG neurons, where PK2 stimulated a large increase in intracellular Ca^++^ release, an effect that was inhibited by Compound 3. Additionally, via immunohistochemical analysis a co-expression of TRPV1 and PKR1, but not PKR2, was detected in small/medium-sized cell bodies of naïve rat DRGs (Figure 10). Subsequently, in vivo experiments were also conducted [50]. Several models (mustard oil (MO), trinitrobenzene sulfonic acid (TNBS), dextran sodium sulfate (DSS), *Citrobacter rodentium* infection (C-ROD), and water avoidance stress (WAS)) were used in mice or rats. In this study, behavioral and biochemical evaluations were performed between 7 and 11 days after IBD induction, therefore not in the acute/initial phase of the pathology. The authors observed that MO mice were characterized by abdominal allodynia, while TNBS and WAS mice were characterized by visceral hyperalgesia (DSS and C-ROD mice were not evaluated for pain-like behavior). For all models, except WAS, a significant increase in PK2 expression levels in the colon was found. This increase was positively correlated with increased mRNA levels of the pro-inflammatory cytokine IL-1β. However, PK2 and PKR1 levels were only overexpressed in the TNBS model. Acute treatment with a specific prokineticin receptor antagonist (Compound 3) was able to counteract pain in MO, TNBS, and WAS mice. Interestingly, in the MO model and TNBS model the Compound 3 therapeutic treatment was compared with alosetron and diclofenac or morphine and diclofenac, respectively. While alosteron (MO mice) and morphine (TNBS mice) were able to decrease abdominal/visceral hypersensitivity, diclofenac had no effect in either of the two models, unlike the PKS antagonist which always resulted effective. Unfortunately, in this study, a biochemical evaluation to verify whether the therapeutic treatments reduced the expression levels of the PKS and pro-inflammatory cytokines was not conducted. These data suggest that, during an intestinal inflammation state, a significant PK2 increase can modulate pain perception thanks to PKRs’ activation at the extrinsic sensory neuron level. PK2, released by resident immune cells and recalled in loco, could then drive visceral inflammatory pain by acting: i. as a chemoattractant for monocytes/macrophages; ii. as a stimulator for the release of pro-inflammatory cytokines; and iii. directly on sensory neurons by sensitizing TRPV1s (Figure 11). 

Additionally, the ineffectiveness of diclofenac in relieving IBD pain suggests that, in an advanced state of the pathology, an acute inhibition of inflammation is not sufficient to provide analgesia. This suggests, in IBD pain, the importance of a direct role of PK2 on PKRs expressed via both extrinsic sensory and enteric neurons innervating the GI tract. Also Zinni et al. [104] evaluated PK2 and PKRs’ expression in a preclinical IBD model (TNBS). This research group focused on the early stage of the disease (4 days post-TNBS) by observing, in the colon, a clear PK2 upregulation that correlates with a significant increase in the pro-inflammatory cytokines IL-1β and TNFα. However, at this time of observation, neither PKR1 nor PKR2 were upregulated, although a positive trend was shown for PKR2. Recently, we performed [39], in a TNBS mouse model, a detailed PKS characterization in different tissues and organs involved in IBDs. In addition to evaluating certain tissues, such as of the mesenteric lymph nodes, colon, myenteric plexus, and DRGs, we were the first to analyze the expression of the PKS at the spinal cord level. Our data, for the first time, suggest that PKS activation is involved in the pathways of visceral pain transmission in IBDs. 

We conducted behavioral assessments up to day 14 from TNBS and performed biochemical assessments at two time points (days 7 and 14) in order to verify the PKS’s activation and the (neuro)inflammatory state both in an early stage and in a more advanced stage of the disease. Our data showed that TNBS animals were characterized by a significant increase in the disease activity index in parallel to the development of abdominal mechanical allodynia, visceral hyperalgesia, and muscle strength decrease. At the biochemical level, our data demonstrated that in both peripheral (myenteric plexus and DRGs) and central (spinal cord) nervous tissues a significant upregulation in the PKS was present and associated with a clear (neuro)inflammatory condition with microgliosis and astrogliosis (Figure 12), as well as high levels of pro-inflammatory cytokines. 

From all these works, it clearly appears that the activation, at different levels, of PKS members could play a fundamental role in the pain and neuroinflammatory components associated with IBDs. On the basis of the literature, we might speculate that PK1/PKR1 plays an important role in the activation of a local inflammatory response, PK2/PKR1 in maintaining this inflammatory condition, as well as in promoting inflammatory pain, and PK2/PKR2 in central sensitization and therefore in pain chronicization. However, further studies have yet to be undertaken to determine the molecular mechanisms underlying these phenomena.

### 4.5. Evidence of the PKS in IBD Patients

To date, clinical research on the PKS in IBDs is still very scarce. Watson and colleagues [52] were the first to evaluate PK2 levels in colonic mucosal biopsies of IBD patients. These authors described that in samples of IBD patients, the PK2 expression levels were significantly increased (approximately 15-fold) compared to healthy samples. They also observed a positive correlation between increased levels of PK2 and the pro-inflammatory cytokine IL-1β. These data suggest the existence, also in humans, of a clear relationship between the PKS and the inflammatory state. Subsequently, two recent studies focused on the identification of possible IBD biomarkers [105,106]. A meta-analysis of expression profiling was performed from white blood cells and colonic biopsies of IBD pediatric patients [105]. This analysis identified that inflammatory mechanisms in peripheral blood only partially reflected those in colon tissue. In detail, only three genes always overlapped in both colon biopsies and white blood cells, i.e., *mmp9* (matrix metallopeptidase 9), *mzb1* (marginal zone B and B1 cell-specific protein), and *pk2*. Subsequently, a further meta-analysis study was conducted [106]. The amount of data processed from several cohorts of IBD patients was the largest ever used to date, and three machine learning algorithms were applied to determine the common feature genes among all the studied cohorts. From the analysis, four characteristic genes showed higher diagnostic value in multiple cohorts, namely *aqp9* (aquaporin-9), *lcn2* (lipocalin-2), *nampt* (nicotinamide phosphoribosyltransferase), and *pk2*. It is interesting to note that in the two independent studies, the only gene in common is *pk2*. This suggests, also in humans, that the PKS is highly connected with IBDs, and that it could prove to be not only a biomarker for diagnostic purposes, but also an effective indicator of disease risk or improvement.

Although clinical studies on the PKS and IBDs are limited, the data shown so far are very interesting, and also indicate a clear involvement/alteration of the PKS in IBD patients.

## 5. Conclusions

From the data reported in the present review, it emerges that the PKS is involved in the regulation of multiple functions, such as gastrointestinal activity, angiogenesis, pain, and tumorigenesis, which are central in IBD development. Moreover, in preclinical models of IBDs, an overexpression of PKS ligands and receptors has been constantly observed in several tissues involved in IBD pathogenesis, suggesting an important role of the PKS in these pathological conditions. Although still limited, clinical evidence has also identified the presence of PKS alterations in tissues obtained from IBD patients. In-depth studies using pharmacological or genetic approaches to manipulate the PKS are needed in order to better understand its possible role in the onset, maintenance, and eventually resolution of IBDs and related pain. Shared efforts must be performed to link preclinical and clinical research together, in order to recognize the PKs as a druggable target to ameliorate IBD patients’ quality of life. 

## Figures and Tables

**Figure 1 biomedicines-11-02985-f001:**
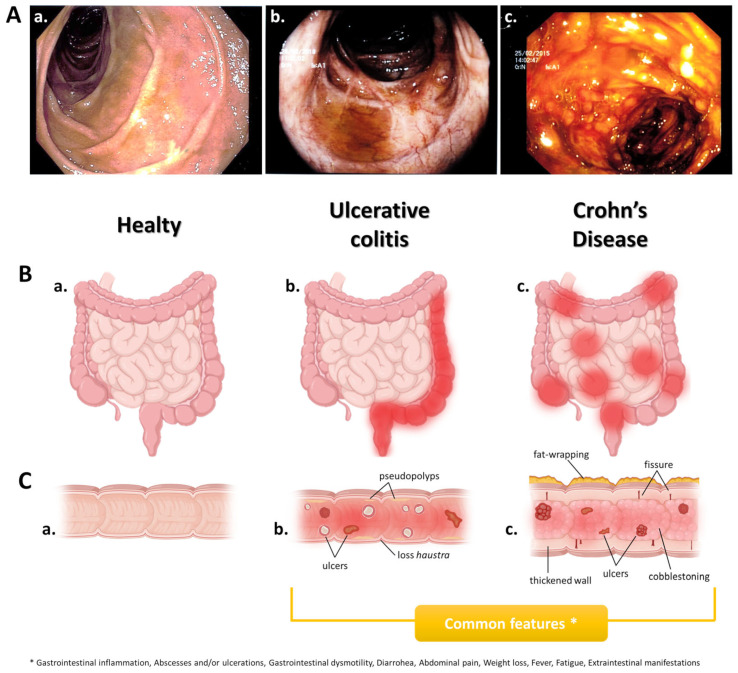
**IBDs: Ulcerative colitis vs. Crohn’s disease**. (**A**) Colonic endoscopy. (**B**) Illustrative representation of the intestine; red indicates the possible inflammation site(s). (**C**) Illustrative representation of the colon longitudinal section and main pathophysiological features. (**a**) Healthy people, (**b**) patients with ulcerative colitis, and (**c**) patients with Crohn’s disease. Drawn by authors using BioRENDER online software.

**Figure 2 biomedicines-11-02985-f002:**
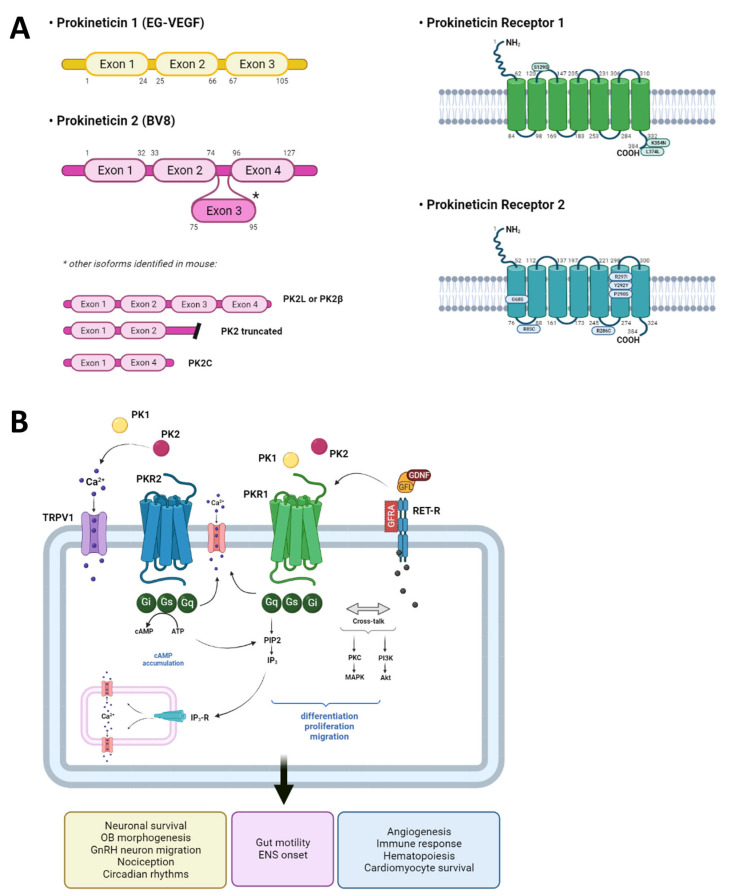
**PKS’s structure and signaling.** (**A**) In mice, the PK1 gene maps to chromosome 3 and the PK2 gene maps to chromosome 6 (in humans, chromosomes 1p13.1 and 3p21.1, respectively). In both mouse and human, the PK1 gene has three exons encoding the mature protein (mouse, 105 amino acids (aa); human, 86 aa), while the PK2 gene has four exons, of which three exons encode the classical mature protein PK2 (mouse, 108 aa; human, 81 aa). Furthermore, in mice, it has been recently identified that via alternative splicing the PK2 gene can also originate other PK2 isoforms (PK2L or PK2beta, 129 aa, is encoded by all four exons; truncated PK2, 74 aa, is encoded by exons 1 and 2 and part of intron 2; and PK2C, 63 aa, is encoded by exons 1 and 4). (**B**) Both PKs can bind to their G protein-coupled receptors, PKR1 and PKR2, activating Gi (starting the MAPK/Akt cascade), Gs (promoting cAMP accumulation), or Gq (inducing calcium mobilization) and their downstream pathways. Drawn by the authors using BioRENDER online software.

**Figure 3 biomedicines-11-02985-f003:**
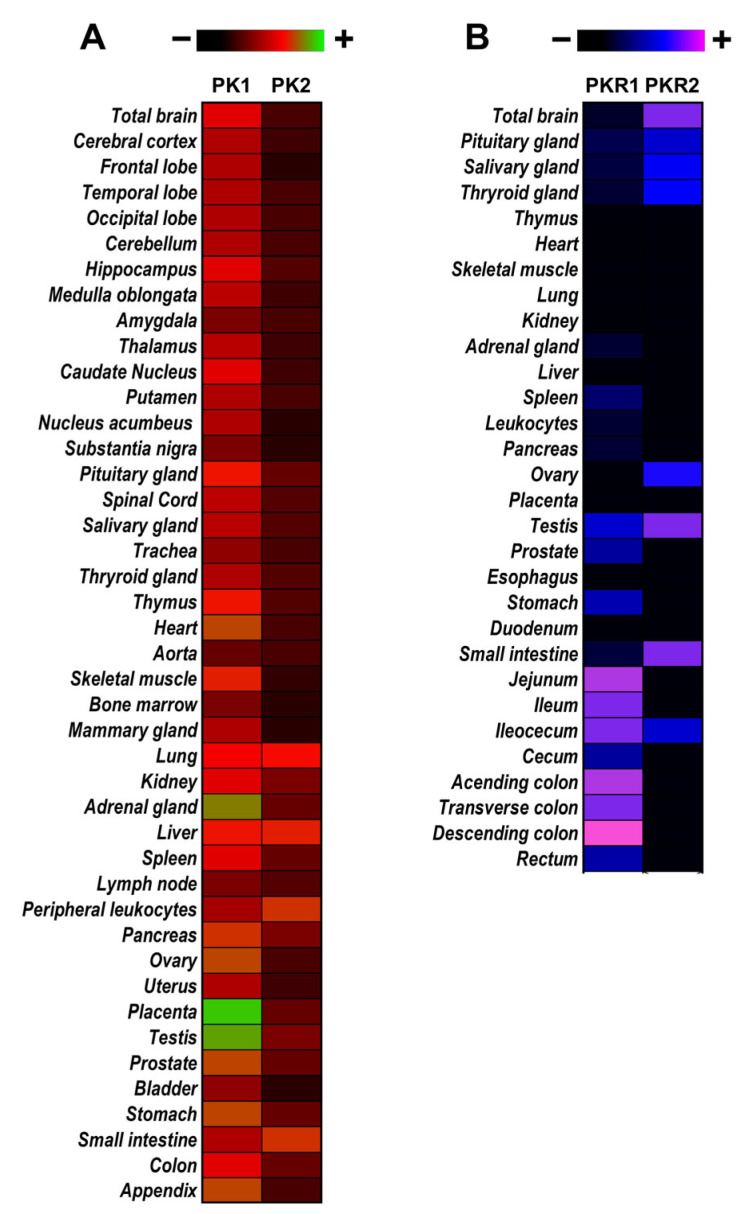
**Expression patterns of the prokineticin system.** Illustrative representation of the expression of the (**A**) PKs [22] and (**B**) PKRs [27] in human tissues.

**Figure 4 biomedicines-11-02985-f004:**
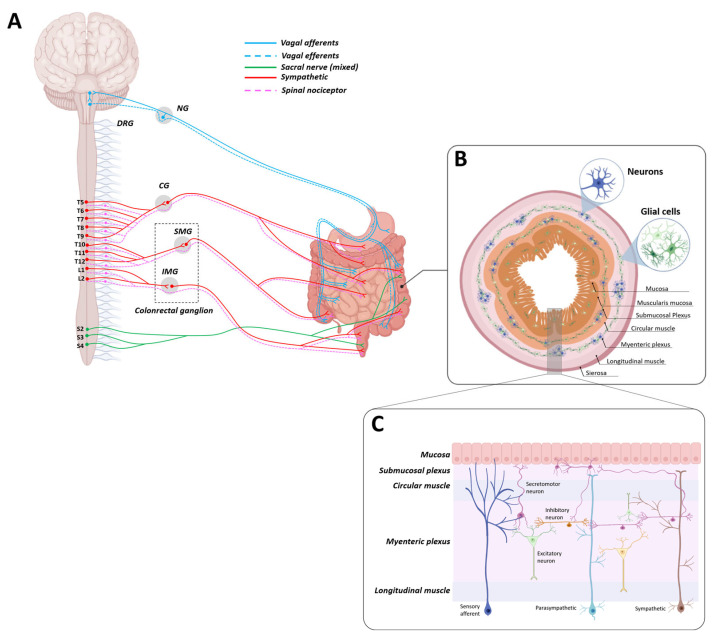
**Neuronal connectome between the gut and the central nervous system (CNS).** (**A**) A schematic representation of the CNS–gut connection. The intestine is connected to the CNS, both through interactions with the enteric nervous system (ENS) and independently through interactions with diverse gastrointestinal cells. Neural pathways that connect the gut to the CNS include the vagus nerve, which consists of the vagal afferent and efferent nerves (in blue, whose neurons reside in the nodose ganglion (NG) and in the brainstem, respectively); the spinal nociceptive nerves (whose neurons reside in the DRG, which are sensory neurons that innervate the viscera and the spinal cord neurons); the postsynaptic sympathetic nerves (whose neurons reside in celiac ganglia (CG) and in superior (SMG) and inferior mesenteric ganglia (IMG)); and the spinal sacral nerve, which directly connects the colon to spinal neurons (sympathetic and parasympathetic components). (**B**) A detailed transversal section of the colon. The gut has its own nervous system (ENS), whose neurons are located in two plexuses, namely the myenteric plexus and submucosal plexus. Neurons of both plexuses innervate different tissue regions, performing several functions; neural pathways of the submucosal plexus mainly regulate fluid exchange across the intestinal mucosa, while those of the myenteric plexus coordinate intestine contractile activity. Additionally, in these intestinal layers, several enteric glial cells are present. (**C**) Illustrative magnification of the intestinal layers, showing neurons and glial cell distribution. Drawn by the authors using BioRENDER online software.

**Figure 6 biomedicines-11-02985-f006:**
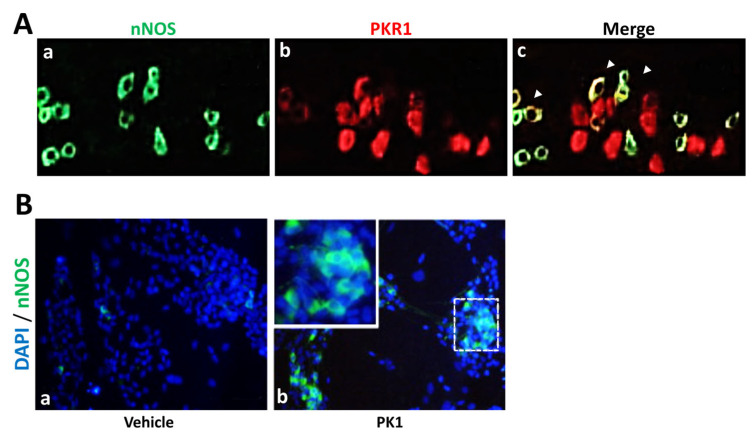
**The PKS and nNOS.** (**A**) Immunofluorescence investigations show that (**a**–**c**) PKR1 co-localizes with neuronal nitric oxide synthase (nNOS) in the mouse musculo-myenteric plexus at the colonic level, and that (**B**) NCCs cultured for 1 week in the presence of (**b**) PK1 display upregulated nNOS expression compared to those cultured with (**a**) vehicle alone. These images have been edited and republished, in accordance with the specific editor copyright, by (**A**) Hoogerwerf [53] and (**B**) Ngan et al. [54].

**Figure 9 biomedicines-11-02985-f009:**
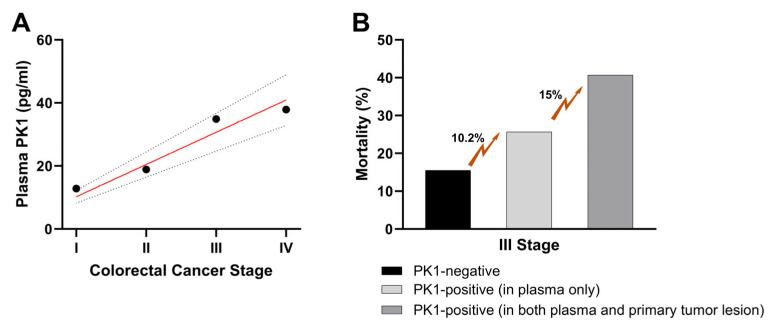
**PK1 expressions in colorectal cancer patients and their prognosis**. (**A**) Plasma PK1 concentration was evaluated in 130 CRC patients; the graph shows an increasing trend of PK1 with the cancer stage progression. (**B**) In CRC patients’ stage III, PK1 levels were evaluated in both plasma and primary tumor lesions (every 10 months, 5-year follow-up). The results were then correlated to the CRC patient mortality rate. From these evaluations, it emerged that in patients who did not express PK1 either in the plasma or at the level of the primary lesion, the mortality was 15.5%, in patients who expressed plasma PK1 the mortality rate was 25.7% (increase of 10.2%), while patients expressing PK1 in both plasma and primary tumors had a mortality of 40.7% (25.2% and 15% increase, respectively). These data have been edited and republished, in accordance with the specific editor copyright, by Tagai et al. [80].

**Figure 10 biomedicines-11-02985-f010:**
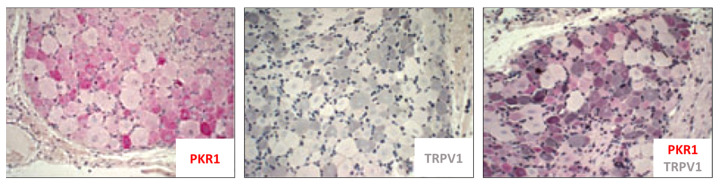
**PKR1 expression in rat DRGs**. Immunostaining in naïve rat DRG shows that PKR1 is not expressed in large-diameter neurons but co-localizes with TRPV1 in the cell bodies of small/medium-diameter nociceptive sensory neurons. These images have been edited and republished, in accordance with the specific editor copyright, by Watson et al. [50].

**Figure 11 biomedicines-11-02985-f011:**
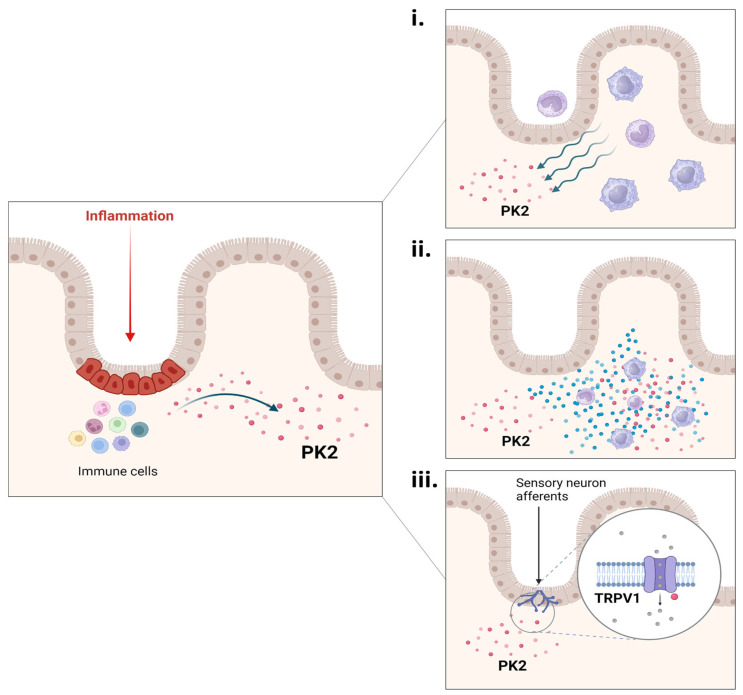
**PK2 increase and visceral pain.** During intestinal inflammation, PK2 is released by immune cells. Its increase at the inflammatory site could modulate/drive visceral pain activating PKRs, which are largely expressed in the gut and enteric nervous system. PK2 can act by: (**i**) attracting monocytes/macrophages to the site of inflammation (chemotactic action); (**ii**) stimulating the release of pro-inflammatory cytokines by immune cells (helping to create a vicious cycle for its production); and (**iii**) by sensitizing TRPV1 localized on the afferents of sensory neurons that innervate the gastrointestinal tract. Drawn by the authors using BioRENDER online software.

**Figure 12 biomedicines-11-02985-f012:**
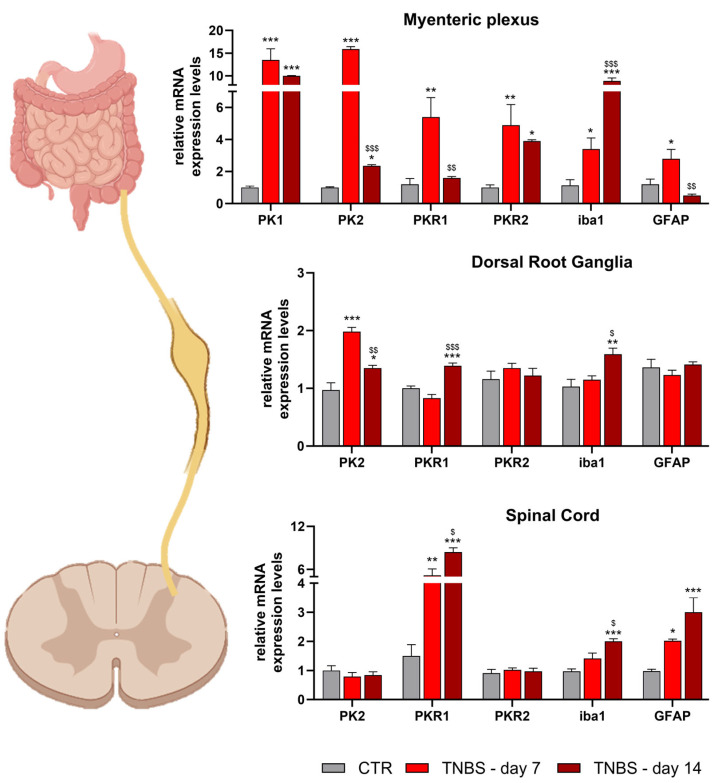
**PKS characterization in nervous tissue of TNBS mice.** PKS and glial marker expression levels in a TNBS murine model. The evaluations were performed in important stations involved in IBD pain transmission, i.e., the myenteric plexus, dorsal root ganglia, and spinal cord, at two different time points (day 7 and day 14 post-TNBS). *** *p* < 0.001, ** *p* < 0.01, and * *p* < 0.05 vs. respective CTR; $$$ *p* < 0.001, $$ *p* < 0.01, and $ < 0.05 vs. TNBS d7. These images have been edited and republished, in accordance with the specific editor copyright, by Amodeo et al. [39].

**Table 1 biomedicines-11-02985-t001:** **The PKS in mammals’ organs and tissues.** The “+” symbol indicates that PKS member(s) were detected as proteins or mRNAs. The “−” symbol indicates that PKS member(s) were not detected. The abbreviation N/A indicates that the data are not available.

	Ligand		Receptor	
Organ/Tissue	PK1	PK2	PKR1	PKR2
** *The brain* **				
Olfactory bulb	N/A	**+**	**+**	**+**
Other olfactory regions	N/A	N/A	N/A	**+**
Olfactory ventricle	N/A	N/A	**+**	**+**
Other ventricles	N/A	N/A	**+**	**+**
Ependyma	N/A	N/A	**+**	**+**
Cerebellum	**+**	**+**	N/A	N/A
Medial preoptic area	N/A	**+**	N/A	N/A
Lateral preoptic area	N/A	N/A	N/A	**+**
Purkinje cells	N/A	**+**	N/A	N/A
Paraventricular nucleus	N/A	N/A	N/A	**+**
Nucleus arcuatus	N/A	**+**	**+**	**+**
Nucleus accumbens	N/A	**+**	N/A	N/A
Ventral pallidum	N/A	N/A	N/A	**+**
Globus pallidus	N/A	N/A	N/A	**+**
Calleja islands	N/A	**+**	N/A	N/A
Suprachiasmatic nucleus	N/A	**+**	N/A	**+**
Amygdala	N/A	**+**	N/A	**+**
Hypothalamus	N/A	**+**	**+**	**+**
Thalamus	N/A	N/A	N/A	**+**
Hippocampus	N/A	**+**	**+**	**+**
Septum	N/A	N/A	N/A	**+**
Dentate gyrus	N/A	N/A	**+**	**+**
Subiculum	N/A	N/A	**+**	**+**
Median eminence	N/A	N/A	N/A	**+**
Mammillary nucleus	N/A	**+**	**+**	**+**
Zona incerta	N/A	N/A	**+**	N/A
Pituitary	N/A	N/A	N/A	**+**
Mesencephalon	N/A	**+**	**+**	**+**
Periacqueductal gray	N/A	N/A	**+**	**+**
Dorsal raphe	N/A	N/A	N/A	**+**
Subfornical organ	N/A	N/A	N/A	**+**
Gasser ganglion	N/A	N/A	**+**	**+**
Cerebral cortex	N/A	**+**	**+**	**+**
Rostral migratory stream	N/A	N/A	**+**	**+**
Brainstem	**+**	**+**	**+**	**+**
Nucleus tractus solitarius	**+**	N/A	N/A	**+**
Dorsal motor nucleus vagus	N/A	N/A	**+**	N/A
Lateral reticular nucleus	**+**	N/A	N/A	N/A
Area postrema	N/A	N/A	N/A	+
** *The nervous system* **				
Spinal cord	N/A	**+**	**+**	**+**
DRGs	N/A	**+**	**+**	**+**
Sciatic nerve	N/A	**+**	**+**	**+**
Myenteric plexus	N/A	**+**	**+**	**+**
Neuronal cells	N/A	**+**	**+**	**+**
Astrocytic cells	N/A	**+**	**+**	**+**
Microglia cells	N/A	−	**+**	**+**
Schwann cells	N/A	+	+	+
** *The gastrointestinal system* **				
Stomach	**+**	**+**	**+**	**+**
Fundus	**+**	**+**	**+**	**+**
Pylorus	**+**	**+**	**+**	**+**
Duodenum	**+**	N/A	**+**	**+**
Jejunum	**+**	N/A	**+**	**+**
Ileum	**+**	**+**	**+**	**+**
Ileum mucosa	**+**	N/A	**+**	**+**
Ileum muscolaris	**+**	**+**	**+**	N/A
Cecum	**+**	**+**	**+**	−
Colon	**+**	**+**	**+**	**+**
Colon muscle layer	N/A	**+**	N/A	**+**
Rectum	**+**	**+**	**+**	−
Submucosal neurons	N/A	N/A	**+**	N/A
Enteric plexus (neurons and fibres)	N/A	N/A	**+**	N/A
Interstitial nerve cells	**+**	N/A	**+**	N/A
Epithelial cells (crypt base)	N/A	N/A	**+**	N/A
Mucosa epithelial cells	**+**	**+**	**+**	**+**
Proximal glands	N/A	**+**	N/A	**+**
Interstitium	N/A	**+**	N/A	**+**
Mesenchyma	+	N/A	+	N/A
** *Immune cells* **				
Hematopoietic stem cells	N/A	**+**	**+**	**+**
Monocytes	−	**+**	**+**	**+**
Neutrophils	N/A	**+**	**+**	**+**
Dentritic cells	N/A	**+**	N/A	N/A
Granulocytes	N/A	**+**	**+**	**+**
Mecrophages	**+**	**+**	**+**	**+**
Lymphocites	**+**	N/A	**+**	**+**
** *Ovary* **				
Granulosa	**+**	N/A	**+**	**+**
Theca cells	**+**	N/A	**+**	**+**
Capillary endothelial cells	N/A	N/A	N/A	N/A
Corpus luteum	+	N/A	+	+
** *Uterus* **				
Grandular epithelium	**+**	**+**	**+**	**+**
Endothelial cells	**+**	**+**	**+**	**+**
Stromal muscle cells	**+**	**+**	**+**	**+**
Smooth muscle cells	**+**	**+**	**+**	**+**
** *Placenta* **	+	N/A	+	−
** *Testis* **				
Leyding cells	**+**	N/A	N/A	N/A
Spermatocytes	N/A	**+**	N/A	N/A
Interstitium endothelial cells	N/A	N/A	**+**	**+**
** *Prostate* **	N/A	N/A	+	−
** *Heart* **				
Cardiovascular tissue	+	N/A	**+**	N/A
Cardiac cells	**+**	N/A	**+**	N/A
** *Kidneys* **				
Epithelial tubules	**+**	N/A	**+**	N/A
Endothelial cells	N/A	N/A	**+**	**+**
** *Adrenal glands* **				
Glomerulosa	**+**	**+**	**+**	**+**
Fasciculate cells	**+**	**+**	**+**	N/A
Endothelial cells	N/A	**+**	**+**	**+**
** *Liver* **				
Kupffer cells	N/A	**+**	**+**	**+**
** *Pancreas* **				
Pancreatic islet	**+**	N/A	N/A	N/A
Stellate cells	**+**	N/A	N/A	N/A
Vascular endothelial cells	N/A	N/A	+	+

## Data Availability

Data sharing not applicable.

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
