# Peer review of "The Prokineticin System in Inflammatory Bowel Diseases: A Clinical and Preclinical Overview"

_biomedicines, 2023, doi:10.3390/biomedicines11112985_

Round 1

Reviewer 1 Report

Comments and Suggestions for Authors

Gilda Amide et al are a narrative overview summarizing the evidenced involvement of Prokineticin System (PKS) in IBD and discuss its potentiality as drug gable target. Recent clinical and preclinical studies report PKS as an emerging therapeutic target for IBDs and alterations are likely to play a significant role in IBDs at multiple levels, as in intestinal motility, local inflammation, ulceration processes, localized abdominal and visceral pain as well as central nervous system (CNS) sensitization leading to development of chronic and widespread pain. The paper is well written and informative to the IBD community. I would like all the figures nrs. 1, 4 and 11 to be clearly labeled from where the images were obtained from e.g. Reproduced with permission from Steed et al., Trend Cell Biol, Elsevier, 2010 [reference].  Figures nr. 5-10, and 12 are OK.

Author Response

Gilda Amide et al are a narrative overview summarizing the evidenced involvement of Prokineticin System (PKS) in IBD and discuss its potentiality as drug gable target. Recent clinical and preclinical studies report PKS as an emerging therapeutic target for IBDs and alterations are likely to play a significant role in IBDs at multiple levels, as in intestinal motility, local inflammation, ulceration processes, localized abdominal and visceral pain as well as central nervous system (CNS) sensitization leading to development of chronic and widespread pain. The paper is well written and informative to the IBD community.

We thank the reviewer for appreciating our manuscript.

I would like all the figures nrs. 1, 4 and 11 to be clearly labeled from where the images were obtained from e.g. Reproduced with permission from Steed et al., Trend Cell Biol, Elsevier, 2010 [reference].  Figures nr. 5-10, and 12 are OK.

There is no reference in figures 1, 2, 4 and 11 because they were drawn by the authors using the bioRENDER online software. We apologize to the reviewer for not reporting this information in the text. Now, this has been reported in the figure legends.

Reviewer 2 Report

Comments and Suggestions for Authors

Dear Editor

 This is an interesting article regarding PKS in IBD. The followings are my comments. The PKS had role in various physiological processes. 

#1.  Table 1, the PKS is detected in variety of organ. How about the level of PKS expression in GI tract compared with another organ?

#2.  Table 1, why "Ileum" and "Ileum mucosa" have different expression of PK2 ligand ?

 #3. Table 1 refers to PKS in mammals. Does Human had similar expression compared with other mammals?

#4. Page 9, PKS is involved in IBD and dysfunctional gut. Is there evidence of PKS in motility disorder such as Achalasia ? Gastroparesis and IBS (irritable bowel syndrome) to support this point?

#5.  Section 4.2 What the author thinks regarding the dysregulation of PKS as the cause or the etiology for IBD?

#6. Section 4.5. As IBD had different ethnic prevalence, i.e., the incidence of IBD is higher in the Western population then the Eastern population, is there any genetic factor leads to PKS expression in different ethnic population? 

Comments on the Quality of English Language

The English is fine for read. 

Author Response

Dear Editor

This is an interesting article regarding PKS in IBD.

We thank the reviewer for appreciating our manuscript.

The followings are my comments. The PKS had role in various physiological processes. 

#1.  Table 1, the PKS is detected in variety of organ. How about the level of PKS expression in GI tract compared with another organ?

PKS is expressed by many different organs. It is difficult to identify a quantitative difference among tissues, since the analysis of PKS expression in the different tissues and organs has often been performed by different authors using different methodologies (mRNA, immunohistochemistry, western blotting etc.). A comprehensive multiorgan comparative study, where it is possible to verify the quantitative difference in PKS levels in different organs, has only been conducted by Zhou's group [references 22 and 27]. We tried to reproduce the characterizations conducted by this group through Figure 3. From these data it is possible to observe that PKS (in particular PK1, PK2 and PKR1) is abundantly expressed under physiological conditions in human GI. The difference with other organs depends on the organ which it is compared with. For example, if we take the lung as reference, we observe that its expression of PK1 and PK2 is similar to the GI one, but lacks the expression of both receptors; similarly to what also happens for the kidney. We can also add that in the study that we performed [reference 39], measuring mRNA levels, PKS expression seemed similar in colon and the nervous system (myenteric plexus, dorsal root ganglia and spinal cord), further showing the relevance of the PKS in both inflammation and pain processing. However, after IBD induction a significant upregulation is observed in all tissues. 

A few sentences have been added on this topic.

Page 6, lines 157-169.

“However, to date, a comprehensive multiorgan comparative study, in which it is possible to verify the quantitative difference in PKS levels in different organs, has only been conducted by Zhou's group [22, 27]; therefore, it still remains difficult to identify a quantitative differences between tissues, since the PKS expression analysis in different tissues and organs has often been performed by different authors with different methodologies (mRNA, immunohistochemistry, western blotting, etc.). However, we can state, on the basis of our recent study [39], that the mRNA levels of PKS in colon and in nervous system (myenteric plexus, dorsal root ganglia and spinal cord) were similar, demonstrating the relevance of PKS in both inflammation and pain processing. We believe that detailed multiorgan characterization studies should be conducted to offer a comprehensive overview of PKS expression. Furthermore, performing a characterization of PKS in different mammals would offer a clearer distribution of PKS making also  the possibility to perform quantitative comparisons between the different species. This could be an important step for translating the results obtained in preclinical studies into the clinic.”

#2.  Table 1, why "Ileum" and "Ileum mucosa" have different expression of PK2 ligand?

In Table 1 the “+” symbol has been assigned based on all the information collected from the literature. In this specific case, PK2 expression was observed in the ileum, while PK2 analysis was not conducted in the ileal mucosa (for this reason the “+” symbol is missing). However, it could be hypothesized that PK2 is also expressed in the ileal mucosa, but this has never been verified. We apologize to the reviewer for not having clearly specified that the absence of “+” symbol could be due both to the actual lack of expression in that given tissue, organ and cell type, as well as a lack of data in this regard. Now the two cases, absence of expression (indicated by the symbol “-“) or of investigation (indicated by the acronym N/A) are reported in the Table 1 and in the corresponding legend.

Page 6, lines 170-171.

“The “+” symbol indicates that PKS member(s) was detected as proteins or mRNAs. The "-" symbol indicates that PKS member(s) was not detected. The abbreviation N/A indicates that the data is not available.”

 #3. Table 1 refers to PKS in mammals. Does Human had similar expression compared with other mammals?

This is an interesting question. In general, when PKS was identified, for example, in rodent tissues/organs, its expression was also found in the corresponding human organs/tissues (e.g. macrophages, gastrointestinal tract, synovial fluid, etc.). Therefore, based on the several studies conducted, it is possible to hypothesize that when PKS is present in the tissues/organs of rodents, it is also present in humans. However, as previously reported, it may be difficult to give a quantitative analysis of the amount of PKS expression in the different species.

A sentence is added.

Page 6, lines 166-169.

“Furthermore, performing a characterization of PKS in different mammals would offer a clearer distribution of PKS making also  the possibility to perform quantitative comparisons between the different species. This could be an important step for translating the results obtained in preclinical studies into the clinic.”

#4. Page 9, PKS is involved in IBD and dysfunctional gut. Is there evidence of PKS in motility disorder such as Achalasia? Gastroparesis and IBS (irritable bowel syndrome) to support this point?

We agree with the reviewer. Evidence of PKS involvement in motility disorders (such as achalasia), gastroparesis or IBSs would lend greater robustness to what we have discussed about the role of PKS in IBDs and related intestinal dysfunction. However, in the literature, there are no studies that investigate this aspect. Although this is a point that is not yet entirely clear, further studies could clarify this aspect. Indeed, we hope that our review can offer to readers new ideas for more in-depth analyses.

#5.  Section 4.2 What the author thinks regarding the dysregulation of PKS as the cause or the etiology for IBD?

The exact etiology of IBDs still remains unknown, but with this review we wanted to highlight the deep involvement of PKS in these pathologies, even if many points still remain to be clarified. However, our idea is that a small inflammation of the colon (that we all experience throughout our lives), in susceptible individuals, can lead to early damage of the colon. This could be related to infectious agents, chemical compounds or metabolic alterations (perhaps related to diet-mediated dysbiosis). The disease’s perpetuation may be due to a failure in resolving the inflammatory response. In this initial inflammation a significant increase/exacerbation of PKS could be fundamental for the establishment of the disease. Initially, a key role could be played by PK1 (which promotes the recall of immune cells and monocyte/macrophage lineage differentiation in situ). Subsequently, the recall of immune cells from different body districts also leads to a significant increase in PK2, which further increases the recruitment of macrophages and promotes their proliferation, thus supporting inflammation (through the production of proinflammatory cytokines and the reduction of anti-inflammatory cytokines) and the local inflammatory pain. Therefore, we believe that PKS activation could be a trigger for the onset of IBDs. Obviously, this is our thinking, and detailed studies are needed to confirm our theory.

#6. Section 4.5. As IBD had different ethnic prevalence, i.e., the incidence of IBD is higher in the Western population then the Eastern population, is there any genetic factor leads to PKS expression in different ethnic population? 

We thank the reviewer for this question, which is very interesting and piqued our curiosity. However, there are no publications that report whether genetic factors lead to different expression of PKS on an ethnic basis, or whether PKS is expressed differently in different ethnic populations. However, PK2 has also been studied and reported in the Chinese population in pathological contexts [Yu et al., 2022 - PMID: 34582411; Wang et al., 2021 - PMID: 33883996]. Therefore, we cannot really possible correlate the degree of PKS expression with IBD incidence in the different ethnic populations. We believe that lifestyle has a great importance in the onset of IBD and this is an aspect that should be further evaluated.

The English is fine for read.

We thank the reviewer for the comment.

Round 2

Reviewer 2 Report

Comments and Suggestions for Authors

The authors response questions well. I have no more questions.

Comments on the Quality of English Language

The English is good to read.